# Efficacy and Safety of Hypofractionated Preoperative Radiotherapy for Primary Locally Advanced Soft Tissue Sarcomas of Limbs or Trunk Wall

**DOI:** 10.3390/cancers13122981

**Published:** 2021-06-14

**Authors:** Hanna Koseła-Paterczyk, Paweł Teterycz, Mateusz J. Spałek, Aneta Borkowska, Anna Zawadzka, Michał Wągrodzki, Anna Szumera-Ciećkiewicz, Tadeusz Morysiński, Tomasz Świtaj, Iwona Ługowska, Patrycja Castaneda-Wysocka, Marcin Zdzienicki, Tomasz Goryń, Piotr Rutkowski

**Affiliations:** 1Department of Soft Tissue/Bone Sarcoma and Melanoma, Maria Sklodowska-Curie National Research Institute of Oncology, 02-781 Warsaw, Poland; pawel.teterycz@pib-nio.pl (P.T.); mateusz.spalek@pib-nio.pl (M.J.S.); aneta.borkowska@pib-nio.pl (A.B.); Tadeusz.Morysinski@pib-nio.pl (T.M.); tomasz.switaj@pib-nio.pl (T.Ś.); iwona.lugowska@pib-nio.pl (I.Ł.); Marcin.Zdzienicki@pib-nio.pl (M.Z.); tomasz.goryn@pib-nio.pl (T.G.); piotr.rutkowski@pib-nio.pl (P.R.); 2Department of Computational Oncology, Maria Sklodowska-Curie National Research Institute of Oncology, 02-781 Warsaw, Poland; 3Medical Physics Department, Maria Sklodowska-Curie National Research Institute of Oncology, 02-781 Warsaw, Poland; Anna.Zawadzka.Fizyk@pib-nio.pl; 4Department of Pathology and Laboratory Medicine, Maria Sklodowska-Curie National Research Institute of Oncology, 02-781 Warsaw, Poland; michal.wagrodzki@pathologist.cc (M.W.); anna.szumera-cieckiewicz@pib-nio.pl (A.S.-C.); 5Early Phase Clinical Trials Unit, Maria Sklodowska-Curie National Research Institute of Oncology, 02-781 Warsaw, Poland; 6Department of Radiology, Maria Sklodowska-Curie National Research Institute of Oncology, 02-781 Warsaw, Poland; patricia.castaneda-wysocka@pib-nio.pl

**Keywords:** soft tissue sarcoma, preoperative, radiotherapy, local recurrence

## Abstract

**Simple Summary:**

The primary treatment of soft tissue sarcomas (STS) is radical resection. The use of adjuvant radiotherapy showed a significantly decreased incidence of local recurrence. In our previous study, we presented that preoperative hypofractionated radiotherapy is safe and efficient for treating an unselected group of patients with STS. This study aimed to assess the treatment scheme’s use in patients with primary STS treated at one institution. The preoperative radiotherapy (RT) scheme consisted of 5 Gy per fraction for a total dose of 25 Gy. Surgery was performed within 2–4 days from the last day of RT. We included 311 patients in this prospective study. In this group, with a significant percentage of patients with extensive, high-grade STS, hypofractionated preoperative radiotherapy was associated with similar local control compared to published studies dedicated to this population. The early tolerance was good, with a small number of late complications.

**Abstract:**

Background: The use of adjuvant radiotherapy (RT) shows a significantly decreased incidence of local recurrence (LR) in soft tissue sarcomas (STS). This study aimed to assess the treatment scheme’s effect in patients with primary STS treated at one institution. Methods: In this phase 2 trial, 311 patients aged ≥18 years with primary, locally advanced STS of the extremity or trunk wall were assigned to multimodal therapy conducted at one institution. The preoperative RT scheme consisted of 5 Gy per fraction for a total dose of 25 Gy. Surgery was performed within 2–4 days from the last day of RT. The primary endpoint was LR-free survival (LRFS). Adverse events of the treatment were assessed. Results: We included 311 patients with primary locally advanced STS. The median tumor size was 11 cm. In total, 258 patients (83%) had high-grade tumors. In 260 patients (83.6%), clear surgical margins (R0) were obtained. Ninety-six patients (30.8%) had at least one type of treatment adverse event. LR was observed in 13.8% patients. The 5-year overall survival was 63%. Conclusion: In this group, with a significant percentage of patients with extensive, high-grade STS, hypofractionated preoperative RT was associated with good local control and tolerance.

## 1. Introduction

Limb-sparing or conservative surgery is a standard of care for the local treatment of soft tissue sarcomas (STS). For most patients with high- or intermediate-grade lesions, perioperative radiotherapy (RT) is applied for local control of the disease. In low-grade lesions, it may be considered if achieving inappropriate margins is anticipated, especially in larger tumors (stage IB) [1]. These recommendations are based, among others, on the results of two randomized prospective trials: one based on postoperative brachytherapy and another using external beam RT; these both showed significant benefits in terms of local control with the addition of adjuvant treatment to limb-sparing surgery [2,3]. The National Cancer Institute of Canada SR-2 randomized phase III clinical trial suggests that RT, in the treatment of STS, can be sequenced either before or after the surgery, and both treatment methods appear to give comparable results in terms of local control of the disease. Both schemes differ in terms of side effects with more acute adverse events of the preoperative schedule and more long-term adverse events, e.g., fibrosis, in the postoperative scheme [4,5]. The recommended fractionation regimen for preoperative RT is a conventionally fractionated regimen of 50–50.4 Gy in fractions of 1.8–2 Gy [6] with a 5–6 week break between RT completion and surgery [1,6].

Compared to standard fractionation in hypofractionated schemes, the total dose of radiation is split into larger doses given per fraction with fewer fractions in total. This type of fractionation is often employed in palliative settings, for instance, in managing pain in bone metastases. Much evidence regarding the efficacy and safety of hypofractionation also comes from the radical treatment of different types of cancers such as prostate or rectal cancers, in which different schemes of fractionation have been developed and used in everyday practice [7,8,9]. Previously, we published results of a prospective phase II clinical trial with 272 patients diagnosed with locally advanced STS. Preoperative hypofractionated RT 5 × 5 Gy was followed by surgery 3–7 days after RT completion. After a median follow-up of 35 months, the estimated 3-year local control was 81%. Notably, the analyzed group of patients presented larger tumors than those in comparative trials. In the above study, nearly 40% patients were treated after inadequate surgery outside of our center (resection of the local recurrence (LR) or scar) [10]. The current work aimed to provide long-term outcomes of preoperative 5 × 5 Gy in a subgroup of patients with primary STS only who underwent long-term follow-up treatment at one institution—The Maria Skłodowska-Curie National Research Institute of Oncology (MSCNRIO).

## 2. Materials and Methods

We included 311 consecutive patients treated in Maria Sklodowska-Curie National Research Institute of Oncology in the Department of Soft Tissue/Bone Sarcoma and Melanoma, Warsaw, Poland, between February 2010 and December 2017. All consecutive patients were included in this, as mentioned above, prospective clinical trial. The study was approved by the local Bio-Ethics Committee, code number 14/2012. The study is registered on the MSCNRIO website (www.pib-nio.pl, accessed on 13 June 2021). All patients participating in the study provided written informed consent. In current analyses we included adult patients with primary, locally advanced STS localized on the extremities or the trunk wall, histological grade 2 or 3 regardless of the lesion size, or deeply seated (>10 cm) grade 1 STS. All patients had a computed tomography (CT) scan of the thorax to exclude metastatic disease prior to treatment. All pathological diagnoses of STS were confirmed by the Department of Pathology and Laboratory Medicine. During the multidisciplinary team meeting (MDT), therapy was planned with the medical oncologist, surgeon, radiation oncologist, pathologist, and radiologist experienced in treating patients with STS. There was no other selection of patients.

### 2.1. Treatment

Neoadjuvant chemotherapy based on doxorubicin combined with ifosfamide or dacarbazine was administered before RT in selected patients at high risk of distant recurrence, with the diagnosis of chemosensitive subtypes. The RT consisted of 5 Gy per fraction delivered on five consecutive days to a total dose of 25 Gy. After individually chosen immobilization and planning CT, delineation was performed by an experienced radiation oncologist. Another radiation oncologist independently reviewed contours. The gross target volume (GTV) was contoured on planning CT fused with contrast-enhanced MRI and/or diagnostic CT. Clinical target volume (CTV) was created by expanding the GTV 2 cm transversally and 4 cm longitudinally and was reduced at anatomical borders of the tumor spreading (i.e., bones and fascias) unless involved. Planning target volume (PTV) was created by expanding the CTV and adding safety margins (0.7–1 cm). Most patients (*n* = 298, 95.8%) were irradiated with three-dimensional static conformal RT, 12 (3.9%) with intensity-modulated RT (IMRT), and one (0.3%) with volumetric modulated arc therapy. We employed 6 and 15 MV photons. The on-line portal imaging device or cone-beam computed tomography was used every day to control the set-up of the treated volume. Median GTVs, CTVs, and PTVs were as follows: 234 cm^3^ (IQR: 91.3–626.9), 880.8 cm^3^ (IQR: 377.05–1790.55), and 1483.3 cm^3^ (IQR: 830.23–2796.7), respectively. Patients diagnosed with synovial sarcoma (49 cases) received also postoperative chemotherapy.

Surgery was performed within 2–4 days from the last day of RT. Wide local excision of the primary tumor was performed. The tissue flaps were not used. Additional boost RT (30 Gy in 15 fractions on tumor bed plus 2 cm margins) was used in case of R1 surgical margins postoperatively in selected patients based on MDT decisions. Acute and late adverse events were defined with a cut-off of three months. After radical multimodal therapy, all patients were followed up at our institution based on the European Society of Medical Oncology (ESMO) recommendations [1].

### 2.2. Statistical Analysis

It was a prospectively designed feasibility study (nonrandomized exploratory clinical trial) with descriptive analysis of the rate of serious wound complications defined similarly to assumptions taken in the randomized phase III clinical trial conducted by the National Cancer Institute of Canada. The assumption of the study was that it would be terminated earlier in the case of unacceptable toxicity of treatment defined as the frequency of occurrence of toxicity ≥grade 2 according to CTCAE in over 40% of the treated patients. The estimated number of patients to be included, based on this exploratory primary endpoint, was not less than 35 patients. The co-primary endpoint was local relapse survival (LRFS). All analyses were performed in the R language environment version 3.5.1 (The R Foundation for Statistical Computing). Patients’ demographics, tumor characteristics, treatment details, and tumor response were analyzed descriptively. The median follow-up estimated with the reverse Kaplan–Meier method was 57.3 months (95% CI 55.1–61.0). The primary objective was to assess the LRFS. The secondary objectives were overall survival (OS), disease-free survival (DFS), distant metastasis-free survival (DMFS), and rate of early and late complications of the treatment. LRFS, OS, DFS, and DMFS were estimated according to the Kaplan–Meier method. LRFS time was calculated from the date of the start of preoperative RT to the date of the most recent follow-up (censored) or LR. DFS time was calculated from the date of the start of preoperative RT to the date of the most recent follow-up (censored), recurrence, or death. DMFS time was calculated from the date of the start of preoperative RT to the date of the most recent follow-up (censored), distant recurrence, or death. OS time was calculated from date of the start of RT to the most recent follow-up (censored) or death.

In multivariate analysis of the factors associated with LRFS or OS, we used Cox proportional hazards models including covariates with *p* < 0.10 in univariate analysis and prespecified variables regarding the treatment (i.e., neoadjuvant chemotherapy, RT boost, and surgical margin). The proportional hazard assumption was assessed using Schoenfeld residuals [11]. In cases where the assumption was not fulfilled, a model with a step function for the hazard was fitted.

All cases were scored using the SARCULATOR app (www.sarculator.com, accessed on 13 June 2021) based on a nomogram for extremity sarcomas. The results were then compared with actual survival rates.

To assess the possible influence of selected factors impacting the side effects of therapy, we used Fisher’s exact test. The differences were considered statistically significant if the *p*-values were <0.05. No *p*-value adjustment for multiple comparisons was applied.

## 3. Results

### 3.1. Patient and Tumor Characteristics

The patient characteristics are presented in Table 1. The median age was 57 years (range 18–91). The median maximal tumor dimension was 10 cm (range 2–31). In most patients, the tumors were localized on the lower limbs (224; 72% of all lesions), with the vast majority on the thigh (160; 71% of all lower limb lesions). In total, 30% of patients received preoperative chemotherapy—Table 2 shows the characteristics of patients with or without chemotherapy. All patients completed the planned treatment and underwent surgery with curative intent.

### 3.2. Local Recurrences

In 260 patients (84%), microscopically negative surgical margins (R0) were obtained. A total of 18 patients received a postoperative boost due to positive surgical margins. LR was found in 43 (13.8%) of the patients. The 5-year LRFS was 81%. The median of LRFS was not reached. The median time from surgery to LR was 14.7 months. In 56% of patients with LR, another limb-sparing radical surgery could be performed; 19% of those patients underwent amputation. In eight cases, patients received systemic therapy only due to synchronous metastatic disease, and three cases were deemed locally non-operable. Factors that had a significant impact on LRFS were histological subtype (*p* = 0.017), resection margins (0.01) (Figure 1), and tumor stage according to TNM classification by AJCC 8th edition [12] (0.04) (Table 1 and Table 3). In multivariate analysis, factors having a significantly worse impact on LRFS were diagnosis of malignant peripheral nerve sheet tumor (MPNST) and leiomyosarcoma (LMS) vs. other histological subtypes (<0.001) and inadequate surgical margins; however, this factor was an independent negative factor only in the first 12 months from the primary surgery (*p* = 0.022) (Table 4).

### 3.3. Survival

A total of 156 patients had disease recurrence at the time of analysis. The 5-year DFS was 46%. Median DFS was 38.6 months (95% CI 22.5–88.5). Factors having a significant impact on DFS were tumor size (*p* < 0.0001) and tumor grade (*p* = 0.0003) (Table 1). Distant metastases were observed in 136 (43.7%) patients, largely to the lungs. The 5-year DMFS was 54% (95% CI 48–60%). The estimated median DMFS was 75.1 months (95% CI 42.6: not reached). In total, 107 patients were dead at the time of the analysis, with the disease’s progression being the most common cause of death. Eleven patients died from other causes not related to their primary diagnosis. The 5-year overall survival rate was 63%. The estimated median OS was 90.4 months (95% CI 73.8-not reached). Factors having a significant impact on OS were age (*p* = 0.025), tumor size (*p* < 0.0001) and tumor grade (*p* = 0.0047) (Table 1 and Table 2, Figure 1). In multivariate analysis, factors having a significantly worse impact on OS were age older than 60 years (*p* = 0.004), higher tumor grade (*p* = 0.001), and larger tumor size (*p* < 0.001) (Table 4.)

### 3.4. Sarculator Comparison

A total of 287 cases were evaluable by the Sarculator app. Among them, 151 had predicted a 10-year OS rate higher than 60% (“good prognosis” group) and 136 had predicted a 10-year OS rate of lower than 60% (“worse prognosis” group). The median predicted that the five-year OS for the whole group was 72% (IQR: 60.5–83.5%), whereas the observed five-year OS rate was 63% (95% CI: 57–70%). The median predicted that the five-year OS in the “good prognosis” group was 83% (IQR: 78–88%) with an observed 5-year OS rate of 83% (95% CI: 77–91%). In the “worse prognosis” group, the median predicted that the 5-year OS rate was 59% (IQR: 53–65%); the observed value reached 40% (95% CI: 32–51%).

### 3.5. Treatment Adverse Events

In total, 96 (30.8%) of the patients had experienced any combined treatment-related adverse events in the treated group. A total of 7.3% (23) of them required surgical management of wound complications, usually a secondary closure. No amputations were required due to wound complications. A higher risk of complications was found in patients with tumors localized to the lower limbs. We did not find a significant increase in adverse events in patients with tumors located in the adductor compartment compared to other lower limb locations. Patients’ age did not influence the adverse events related to wound complications. Distribution and factors correlated with the onset of adverse events are shown in Table 5.

### 3.6. Early Adverse Events (Occurred within <3 Months after Surgery)

Most of the complications were acute and reversible and diagnosed in 75 (24%) treated patients. In total, 51 (16.3%) patients had prolonged healing of the wound (>1 month), 29 (9.3%) had wound dehiscence, wound infection requiring oral antibiotics concerned 25 (8%) of patients. Distribution and factors correlated with the early onset of adverse events are shown in Table 5.

### 3.7. Late Adverse Events (>3 Months after Surgery)

The treatment’s late complications were less common and occurred in 27 (8.6%) of the patients. The most common in this group was prolonged edema of the operated limb 8 (2.5%). One patient had severe fibrosis, leading to limb contracture. Three patients had a fracture of the treated limb, all of whom had tumors localized on the lower limbs.

## 4. Discussion

This prospective study shows the results of a large series of patients with primary locally advanced STS treated with preoperative hypofractionated RT. It showed that this combined treatment modality is feasible, safe, and efficient, without significantly increased adverse events of the treatment as compared to previously published reports with conventionally fractionated preoperative RT.

The rationale for hypofractionation regimen was based on radiobiological grounds. Data is limited in terms of determination of alpha/beta ratio in sarcomas. Based on the results for liposarcomas published by Reitan and Kaalhus the a/b ratio was estimated at 0.4 (−1.4, 5.4) [13,14]. To compare the total doses given to the patients with the use of various fraction doses in various treatment periods, we converted the physical doses into biologically equivalent doses (BEDs). The following formula proposed by Fowler [15] and modified for easy daily practice calculations was used: BED = nd_applied_ (d_applied_ + a/b)/(d_reference_ + a/b) where *n* is the number of fractions, applied is the fraction size of the applied regimen, d_reference_ is the conventional fraction size of 2 Gy, a/b is the ratio of radiation fractionation sensitivity (which has been assumed to be equal to 0.4 Gy for liposarcomas according to Reitan and Kaalhus). Using this formula, the BED was 56.25 Gy for 5 × 5 Gy scheme compared to 50 Gy in conventional schedule of 25 × 2 Gy and 33 total treatment days. It was described earlier in our papers.

The patients treated in our study consisted of patients who, in the vast majority, had a high risk of disease recurrence (about 85% of a high-grade tumor, larger than 5 cm); in this patient group, surgery alone would not be sufficient treatment [16]. In our analyzed group, the factors harming local control of the disease were histological subtype (MPNST and LMS vs. others) and positive resection margins, similar to results obtained in previously published studies [17,18]. Notably, the negative impact of inadequate resection margins diminishes after the first 12 months after surgery, which could be explained by the less aggressive biologic character of some tumors. We also analyzed our large, heterogeneous group of patients in terms of prognosis according to TNM classification by AJCC 8th edition [12], confirming the poor prognosis of patients with large, high-grade tumors in terms of both high risks of local failure and death due to disease progression. When comparing LR rates, our group’s results were consistent with those reported in other studies and local control of the disease exceeded 80% in the 5 years [4,19,20].

In the multivariable model, we did not show differences between patients who received chemotherapy and those who did not. This may be due to several factors. First, our research was focused on radiotherapy treatment. This study was neither designed nor powered to show differences in survival in those two groups. Secondly, as we showed in Table 3, the groups with and without chemotherapy were not balanced. There may be a residual confounding present in the multivariable models, which influences the results. Lastly, this study extends over eight years. At this time, several chemotherapy regimens were used. Possibly, not all of them were optimal and visibly affected OS of the patients. At the same time, the HR (0.85) coefficient in the model in Table 4 suggests benefit in chemotherapy patients and a 15% reduction in risk of death. The effect is not significant due to the factors mentioned earlier. Growing evidence highlights the efficacy of increasing patients’ chance for survival using adjuvant chemotherapy in groups with the worst prognosis. Pasquali et al. used the prognostic nomogram, Sarculator, to analyze patients’ survival in the EORTC-STBSG 62931 randomized trial, which failed to detect an impact of adjuvant doxorubicin plus ifosfamide over observation in preoperative STS treatment. These analyses showed that adjuvant chemotherapy decreased the risk of recurrence and death by half in the group of patients with a low predicted probability of overall survival [18]. It is worth noting the high percentage of patients in our group with worse prognosis when assessed using the prognostic nomogram Sarculator. In this group of 136 patients, 44 (32%) received adjuvant chemotherapy, and 92 (67%) did not receive perioperative systemic treatment. In the group of patients with better survival, when assessed by Sarculator, but who, according to ESMO guidelines, were candidates for chemotherapy, preoperative systemic therapy was given to 30% of patients [1].

One must notice that the results seem worse than those noted in the preoperative RT trials using more modern RT techniques such as IMRT. When Folkert et al. compared data obtained from one institution of patients treated with IMRT or conventional external beam RT, nearly two times lower LR of the disease (7.6 % vs. 15.1% 5-year LR, respectively) was observed in the IMRT group even though the analyzed groups were not well balanced in terms of unfavorable prognostic factors such as positive resection margins [21]. Better results with IMRT in this study can be explained by the better dose conformity [21]. In our group, only 4% patients received preoperative RT with dynamic RT techniques.

The used treatment technique and timing also impact the rate of adverse events of the treatment. In the randomized, phase III clinical trial conducted by the National Cancer Institute of Canada, preoperative RT was associated with a significantly higher risk of early wound complications (35% vs. 17%) when compared to postoperative RT [4]. However, such treatment complications are usually transient compared to late toxicities such as fibrosis or edema [5,22]. In our group, the early complication rate was not high (24%), and only 7% patients required second surgical interventions to treat the adverse events. Like in other studies, in our group too, patients with lesions localized on the lower limbs had a significantly higher risk of complications [23]. Again, the use of older RT techniques is a limitation in our group as IMRT can reduce treatment complications. Studies on IMRT in STS showed a reduced risk of side effects of the treatment (30.5% wound complications in the trial by Sullivan et al., and 10.5% late complications in the study by Wang et al.) and the need for additional surgical procedures when compared to data obtained from the randomized National Cancer Institute of Canada trial [24,25].

The advantages of hypofractionated RT are shorter overall treatment time and probably higher cost-effectiveness ratio, convenience for the treating team and patients, and better reproducibility (fewer fractions). We know that hypofractionated RT, as a part of radical therapy, is safe and efficient based on experience in other tumors such as intermediate-to-high risk prostate cancer, where hypofractionated RT showed non-inferior results to conventionally fractionated RT in terms of failure-free survival, and the late adverse events were similar in both treatment groups [8]. In addition, no differences in oncological outcomes were reported between chemoradiotherapy with standard fractionation and surgery 4–6 weeks later and hypofractionated RT 5 × 5 Gy with surgery within seven days for rectal cancer, with no statistically significant differences in postoperative complications [9]. We also did not note an increased risk of long-term complications in our study, with less than 9% patients experiencing late radiation effects, which could be a concern in such a relatively extreme hypofractionated strategy [26].

Altered preoperative RT regimens in STS have been recently extensively studied [26]. A higher dose was used in a recently published study by Kalbasi et al. [27]. The authors included patients with STS located on the limbs and trunk wall. Among the patients, 74% had tumors bigger than 5 cm; all but one had high-grade lesions. 30 Gy in five fractions on consecutive days were delivered, followed by surgery 2–6 weeks later. In 76% of patients, IMRT was used. Overall, 16% patients developed at least one grade 2 radiation-associated adverse event, and 32% had significant wound complications. With a minimum of 2 years follow-up, 5.7% patients developed LR after surgery, which is an excellent result.

One of the benefits of preoperative RT is the chance of tumor downsizing, thus easing radical resection in the case of large borderline resectable lesions [28]. We also studied the same 5 × 5 scheme but with a much longer interval between RT and surgery. In the first study, which included only patients with a very radiosensitive subtype of STS—myxoid liposarcoma—the wide local excision was performed after a median of 7 weeks (range 5–10 weeks) from the end of RT. In those assessed for tumor response by RECIST criteria, partial responses were noted in 58% patients [29]. Our other study was dedicated to patients with marginally resectable STS, where the hypofractionated RT was used in combination with chemotherapy used in the interval between RT and surgery; responses in terms of decrease in tumor size were observed [30]. This response was not observed in our group, where the break between RT completion and surgery did not exceed one week. This lack of tumor shrinkage may be an issue, especially in the large lesion where size reduction would be much wanted.

## 5. Conclusions

In this study, we presented results of treatment of one of the largest published groups of STS patients treated with hypofractionated preoperative RT so far. The efficacy of the therapy and its adverse events are similar to those observed when standard fractionation is used. Hypofractionation is an innovative way of preoperative treatment of sarcomas, which definitely shortens the whole treatment and makes it comfortable for both the patient and the treating center. Increasing studies are focusing on the use of alternative preoperative RT schemes in STS, and we believe that the 5 × 5 scheme is worth further exploration and use in clinical practice.

## Figures and Tables

**Figure 1 cancers-13-02981-f001:**
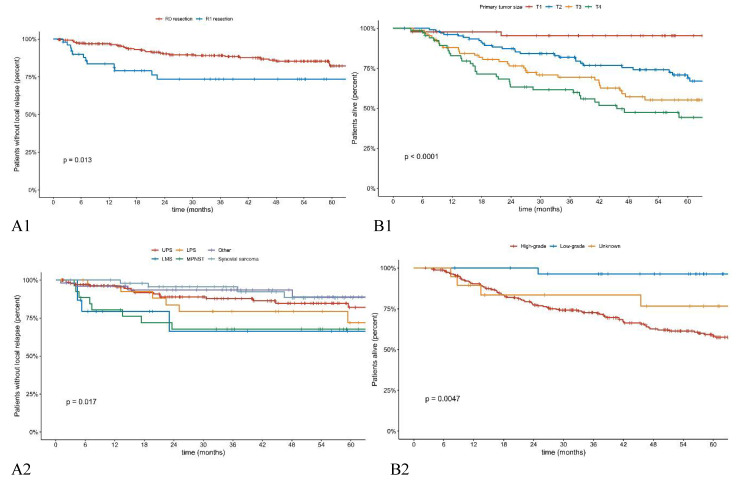
Factors having an impact on LRFS: (**A1**) resection margin, (**A2**) tumor subtype and OS: (**B1**) tumor size, and (**B2**) tumor grade.

**Table 1 cancers-13-02981-t001:** Patient characteristics and univariate analysis of the following factors: LRFS, DFS, and OS.

Patient Characteristics		Overall *n* (%)	LRFS				DFS				OS			
		311	5-Year Survival (%)	95% CI		*p*	5-Year Survival (%)	95% CI		*p*	5-Year Survival (%)	95% CI		*p*
Sex	Female	162 (52.1)	81.4	73.3	90.3	0.4	48.3	40.4	57.7	0.3	67.5	59.6	76.5	0.08
	Male	149 (47.9)	80.6	73.6	88.4		43.4	35.5	53.1		58.5	50.1	68.1	
Age	≤60 years	179 (57.6)	83.1	76.7	90	0.36	44.6	37.4	53.1	0.78	66.2	58.8	74.6	0.025
	>60 years	132 (42.4)	76.6	65.8	89.2		47.6	38.3	59.2		59.2	50.3	69.7	
Subtype	Pleomorphic sarcoma	142 (45.7)	82	73.7	91.2	0.017	42.6	34.4	52.6	0.31	59.2	50.5	69.5	0.29
	Other	50 (16.1)	88.9	78.4	100		56	42.3	74.2		75.7	62.6	91.5	
	Synovial sarcoma	49 (15.8)	88.4	78	100		51	38	68.4		65.2	50.7	83.8	
	Liposarcoma	29 (9.3)	72	54.6	95.1		33.1	18.8	58.1		61	45.3	82.1	
	MPNST	26 (8.4)	67.7	51.5	89		49	32.7	73.2		67.1	50.5	89.2	
	Leiomyosarcoma	15 (4.8)	66.2	42.5	100		55	32.9	91.9		46.4	25.9	83.3	
Grade	Low	30 (9.7)	92	81.7	100	0.15	84.9	72.2	99.9	0.0003	96.3	89.4	100	0.0047
	High	259 (84.1)	78.1	71.5	85.3		40.6	34.4	47.8		58.4	51.8	65.8	
	Unknown	19 (6.2)	94.7	85.2	100		51.8	32.1	83.6		76.5	58.6	100	
Anatomic site of the primary tumor	Trunk	35 (11.3)	72.8	54.3	97.5	0.09	56.4	39.8	79.9	0.33	62.1	46.4	83.2	0.89
	Upper extremity	52 (16.7)	72.6	60	87.8		41.4	28.2	59.9		71.8	59.1	87.2	
	Lower extremity	224 (72)	84.4	78.3	90.9		45.2	38.7	52.9		61.3	54.4	69.1	
Lower extremity tumor location	Adductor compartment	50 (22.3)	73.3	55.8	96.3	0.54	45.2	32.7	62.5	0.84	53.7	39.2	73.7	0.73
	Thigh other	111 (49.6)	88.1	81	95.8		44.1	34.8	56		61	51.5	72.2	
	Lower leg	63 (28.1)	86.7	77.7	96.8		48.4	37.1	63.2		69.5	57.8	83.6	
Maximal size of the tumor	T1 (≤5 cm)	47 (15.2)	88.1	74.9	100	0.1	75.4	60.2	94.5	<0.0001	95.4	89.3	100	<0.0001
	T2 (>5, ≤10 cm)	108 (35)	84.8	77.3	93		49.2	40.1	60.3		69	59.5	79.9	
	T3 (>10, ≤15 cm)	85 (27.5)	76.6	66.7	87.9		35.3	52.5	48.9		55.3	44.5	68.7	
	T4 (>15 cm)	69 (22.3)	72.8	56.3	94.2		34.7	24.4	49.4		44.3	32.5	60.3	
Tumor stage (AJCC 8th edition)	IA	6 (1.9)	100	100	100	0.04	100	100	100	<0.0001	100	100	100	<0.0001
	IB	27 (8.7)	91	80	100		80	66	97		91	81	100	
	II	41 (13.1)	87	74	100		73	57	93		94	87	100	
	IIIA	98 (31.5)	84	76	93		45	36	57		68	58	80	
	IIIB	137 (44)	73	63	84		29	21	29		43	35	54	
	Unknown	2 (0.6)	-	-	-		-	-	-		-	-	-	
Preoperative chemotherapy	YES	94 (30.2)	82.3	74	91.5	0.82	39.9	30.5	52.2	0.09	62	51.3	74.8	0.33
	NO	217 (69.8)	80.4	73.5	88		48.7	41.8	56.8		63.7	56.9	71.3	
Margin status	R0	260 (83.6)	82.3	76.1	88.9	0.01	46.6	40.2	54	0.08	63.5	57.1	70.7	0.38
	R1	51 (16.4)	73.5	61.5	87.8		41.5	29.7	58		60.6	47.7	76.9	
Postoperative radiotherapy (boost)	YES	18 (5.8)	68	48.1	96	0.05	26.6	11.3	62.5	0.06	61.9	41.6	92.1	0.59
	NO	291 (94.2)	82	76.2	88.1		47	41	53.9		63.2	57.2	70	

**Table 2 cancers-13-02981-t002:** Characteristics of patients without or with perioperative chemotherapy.

Patient Characteristics		Without Chemotherapy	With Chemotherapy	*p*
		*n* = 217 (%)	*n* = 94 (%)	
Sex (%)	Female	108 (49.8)	54 (57.4)	0.26
	Male	109 (50.2)	40 (42.6)	
Age (mean (SD))		59.12 (16.55)	46.56 (14.45)	<0.001
Age	≤60	106 (48.8)	73 (77.7)	<0.001
	>60	111 (51.2)	21 (22.3)	
Subtype	Pleomorphic sarcoma	105 (48.4)	37 (39.4)	<0.001
	Leiomyosarcoma	12 (5.5)	3 (3.2)	
	Liposarcoma	26 (12.0)	3 (3.2)	
	MPNST	23 (10.6)	3 (3.2)	
	Other	46 (21.2)	4 (4.3)	
	Synovial sarcoma	5 (2.3)	44 (46.8)	
Tumor grade	High	168 (78.5)	91 (96.8)	<0.001
	Low	30 (14.0)	0 (0.0)	
	Unknown	16 (7.5)	3 (3.2)	
Tumor location	Upper extremity	31 (14.3)	21 (22.3)	0.21
	Trunk wall	25 (11.5)	10 (10.6)	
	Lower extremity	161 (74.2)	63 (67.0)	
Tumor size (mean (SD))		11.43 (6.04)	11.08 (6.35)	0.64
Tumor size	*T1* (≤5 cm)	29 (13.5)	18 (19.1)	0.5
	*T2* (>5, ≤10 cm)	74 (34.4)	34 (36.2)	
	*T3* (>10, ≤15 cm)	63 (29.3)	22 (23.4)	
	*T4* (>15 cm)	49 (22.8)	20 (21.3)	
Resection margin	R0	181 (83.4)	79 (84.0)	1
	R1	36 (16.6)	15 (16.0)	

**Table 3 cancers-13-02981-t003:** Factors affecting the LRFS and OS.

Variable	Patient Characteristics	Univariable HR (95% CI) for LRFS [*p*-Value]	Univariable HR (95% CI) for OS [*p*-Value]
Sex	Female	-	-
	Male	1.29 (0.71–2.35), [*p* = 0.408]	1.41 (0.96–2.06)., [*p* = 0.079]
Age	Mean (SD)	1.02 (1.00–1.04), [*p* = 0.027]	1.02 (1.01–1.03), [*p* < 0.001]
	≤60	-	-
	>60	1.32 (0.72–2.41), [*p* = 0.369]	1.54 (1.05–2.26), [*p* = 0.027]
Subtype	Pleomorphic sarcoma	-	-
	Leiomyosarcoma	2.85 (0.96–8.50), [*p* = 0.060]	1.48 (0.67–3.25), [ *p* = 0.333]
	Liposarcoma	1.55 (0.61–3.93), [*p* = 0.357]	0.81 (0.43–1.52), [*p* = 0.503]
	MPNST	2.57 (1.11–5.95), [*p* = 0.028]	0.89 (0.44–1.81), [*p* = 0.753]
	Other	0.61 (0.21–1.82), [*p* = 0.380]	0.58 (0.30–1.11), [*p* = 0.101]
	Synovial sarcoma	0.59 (0.20–1.74), [*p* = 0.335]	0.66 (0.37–1.18), [*p* = 0.161]
Grade	High	-	-
	Low	0.34 (0.08–1.40), [*p* = 0.135]	0.15 (0.04–0.61), [*p* = 0.008]
	Unknown	0.31 (0.04–2.27), [*p* = 0.249]	0.53 (0.20–1.45), [*p* = 0.218]
Localization	Upper extremity	-	-
	Trunk	0.70 (0.26–1.88), [*p* = 0.483]	1.05 (0.50–2.21), [*p* = 0.890]
	Lower extremity	0.48 (0.24–0.95), [*p* = 0.034]	1.13 (0.67–1.91), [*p* = 0.658]
Size	T1 (≤5 cm)	-	-
	T2 (>5, ≤10 cm)	1.99 (0.57–6.98), [*p* = 0.283]	2.69 (1.04–6.94), [*p* = 0.040]
	T3(>10, ≤15 cm)	3.58 (1.04–12.31), [*p* = 0.043]	4.90 (1.92–12.52), [*p* = 0.001]
	T4 (>15 cm)	3.13 (0.86–11.41), [*p* = 0.083)	6.46 (2.53–16.51), [*p* < 0.001]
Tumor stage (AJCC 8th edition)	I	-	-
	II	1.28 (0.21–7.68); [*p* = 0.785]	1.27 (0.3–5.31); [*p* = 0.74]
	IIIA	2.33 (0.52–10.42); [*p* = 0.268]	3.14 (0.95–10.35); [*p* = 0.06]
	IIIB	4.42(1.04–18.69); [*p* = 0.043]	7.42 (2.33–23.61); [*p* = 0.001]
Margin status	R0	-	-
	R1	2.27 (1.17–4.43), [*p* = 0.016]	1.23 (0.76–1.99), [*p* = 0.390]
Postoperative radiotherapy (boost)	No	-	-
	Yes	2.49 (0.98–6.33), [*p* = 0.056]	1.22 (0.59–2.51), [*p* = 0.591]
Preoperative chemotherapy	No	-	-
	Yes	1.08 (0.57–2.04), [*p* = 0.819]	0.81 (0.53–1.24), [*p* = 0.333]

**Table 4 cancers-13-02981-t004:** Multivariate analysis of variable factors for LRFS and OS (CI—confidence interval).

Local Recurrence-Free Survival			
Patient Characteristics	HR	95% CI	*p*-Value
Age (≤60 years) = ref	-	-	-
Age (>60 years)	1.321	0.687–2.539	0.404
Subtype (MPNST/Leiomyosarcoma vs. other)	3.578	1.797–7.121	<000.1
Grade (high vs. other)	2.824	0.848–9.404	0.091
T1 = ref	-	-	-
T2	1.963	0.553–6.965	0.297
T3	3.461	0.981–12.205	0.054
T4	2.623	0.712–9.66	0.147
Preoperative chemotherapy (Yes vs. No)	1.272	0.635–2.548	0.496
Postoperative radiotherapy (boost)	1.628	0.484–5.475	0.431
Margin R0 up to 1 year of observation	-	-	-
Margin R1 up to 1 year of observation	3.763	1.215–11.657	0.022
Margin R0 more than 1 year of observation	-	-	-
Margin R1 more than 1 year of observation	0.907	0.265–3.099	0.876
**Overall Survival**			
**Patient Characteristics**	**HR**	**95% CI**	***p*-Value**
Sex (male vs. female)	1.424	0.955–2.123	0.083
Age (≤60 years) = ref	-	-	-
Age (>60 years)	1.542	1.019–2.333	0.04
Grade (high vs. other)	4.552	1.831–11.319	0.001
T1 = ref	-	-	-
T2	2.348	0.903–6.108	0.08
T3	4.458	1.723–11.53	0.002
T4	5.755	2.241–14.776	<0.0001
Preoperative chemotherapy (Yes vs. No)	0.853	0.543–1.339	0.489
Postoperative radiotherapy (boost)	1.058	0.43–2.604	0.902
Surgical margin (R1 vs. R0)	1.096	0.598–2.012	0.766

**Table 5 cancers-13-02981-t005:** Treatment adverse events.

Factor N (%)	All Adverse Events		Acute Adverse Events						Late Adverse Events			
			Infection N (%)		Wound Dehiscence N (%)		Prolonged Wound Healing N (%)		Prolonged Edema N (%)		Increased Tissue Fibrosis N (%)	
The anatomic site of the primary tumor												
Trunk (35)	3 (8.6)	<0.001	0	0.016	0	0.004	0	<0.001	1 (2.8)	0.85	0	1
Upper extremity (52)	7(13.4)		1 (1.9)		1 (1.9)		3 (5.7)		0		1 (1.9)	
Lower extremity (224)	86 (38.3)		24 (10.7)		28 (12.5)		48 (21.4)		7 (3)		0	
Preoperative chemotherapy												
Yes (94)	27 (28.7)	0.689	14 (14.8)	0.006	13 (13.8)	0.089	15 (15.9)	1	3 (3)	0.7	0	1
No (217)	69 (31.7)		11 (5)		16 (7.3)		36 (16.5)		5 (2.3)		1 (0.4)	
Postoperative radiotherapy (boost)												
Yes (18)	5 (2.8)	1	0	0.38	1 (5.5)	1	0	0.05	1 (5)	0.38	0	0.3
No (291)	91 (31.2)		25 (8.6)		28 (9.6)		51 (17.5)		7 (2.4)		1	

## Data Availability

Data is contained within the article.

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
