# Peer review of "Efficacy and Safety of Hypofractionated Preoperative Radiotherapy for Primary Locally Advanced Soft Tissue Sarcomas of Limbs or Trunk Wall"

_cancers, 2021, doi:10.3390/cancers13122981_

Round 1

Reviewer 1 Report

First, I have to congratulate the authors for bringing together over 300 patients with STS patients treated with neoadjuvant hypofractionated radiotherapy. This is a remarkable effort.

Apparently, this study was designed earlier than 2010 and recruitment duration with 7 years is quite long. What is missing in the presentation is the definition of the primary endpoint of this prospective clinical trial and the possible secondary end points. These should have been defined in the methods section. Also, for prospective trials it is common and almost mandatory to provide a consort diagram. How many patients were screened? How many patients were lost to follow up?

Regarding the results section the patients characterics table is very detailed, which is good for understanding the population.

However, it should be mentionend wether there were any patient who received postoperative chemotherapy, if any. Currently, only preoperative chemotherapy is listed in the table.

Altogether, this manuscript needs some improvements as listed above but will be worth publishing after that.

Author Response

First, I have to congratulate the authors for bringing together over 300 patients with STS patients treated with neoadjuvant hypofractionated radiotherapy. This is a remarkable effort.

 Thank you for this opinion.

Apparently, this study was designed earlier than 2010 and recruitment duration with 7 years is quite long. What is missing in the presentation is the definition of the primary endpoint of this prospective clinical trial and the possible secondary end points. These should have been defined in the methods section.

Thank you for this remark – it has been added to statistical section. It was prospectively designed feasibility study (nonrandomized exploratory clinical trial) with descriptive analysis the rate of serious wound complications defined similarly to assumptions taken in the randomized phase III clinical trial conducted by the National Cancer Institute of Canada. The assumption of the study was that it would be terminated earlier in the case of unacceptable toxicity of treatment defined as the frequency of occurrence of toxicity ≥grade 2 according to CTCAE in over 40% of the treated patients. The estimated number of patients to be included, based on this exploratory primary endpoint, was not less than 35 patients. The co-primary endpoint was local relapse survival (LRFS).

Also, for prospective trials it is common and almost mandatory to provide a consort diagram. How many patients were screened? How many patients were lost to follow up?

All patients who signed the informed consent and fulfilled the inclusion criteria of the trial were included in the study – there was no other selection or loss of patients at the stage of hypofractionated radiotherapy – it has been added to the manuscript text.

Regarding the results section the patients characterics table is very detailed, which is good for understanding the population.

 Thank you for this comment.

 However, it should be mentionend wether there were any patient who received postoperative chemotherapy, if any. Currently, only preoperative chemotherapy is listed in the table.

Only patients diagnosed with synovial sarcoma (49 cases) received also postoperative chemotherapy. The use of chemotherapy did not impact survival in our group of patients, nor it did not result in higher rates of treatment complications. It has been added to Treatment section.

Altogether, this manuscript needs some improvements as listed above but will be worth publishing after that.

Thank you for this opinion.

Reviewer 2 Report

In this paper the authors describe their experience on the use of preoperative hypofractionated radiotherapy in sarcomas of the limbs and trunk.
Even if the population examined is well represented by 311 cases, the study method adopted cannot be prospective. Ethically I do not understand why a randomized or a phase II similar to that presented in the introduction for conventional RT was not designed. Would a chart review have been better? The doses of RT were chosen on what basis? The answers to these questions should be integrated into the text.
Just as there is a lack of toxicity data that I would expect to see in older patients. Furthermore, how did elderly patients tolerate the therapies? Have I guess the doses been reduced or omitted of doxo for example? Although well presented, the general data are few and little discussed.
I don't understand the difference between the LRFS of 81% at 5 yrs and the DFS 46% at 5 yrs, shouldn't they coincide? Or are they calculated differently?
Several questions should be addressed.

Author Response

In this paper the authors describe their experience on the use of preoperative hypofractionated radiotherapy in sarcomas of the limbs and trunk.

  1. Even if the population examined is well represented by 311 cases, the study method adopted cannot be prospective.

The additional statistical methodology of this prospective trial has been added in Statistical analysis section.

  1. Ethically I do not understand why a randomized or a phase II similar to that presented in the introduction for conventional RT was not designed.

Of course, it would be optimal to have a randomized study comparing our hypofractionated scheme to the conventional one, unfortunately, it is extremely difficult to conduct such a study, as very large number of patients would be required in order to show a statistical significance (in non-inferiority design). Therefore, we decided to conduct smaller, one arm studies based on the 5x5 scheme to have reference data in clinical practice. We mentioned them in the Discussion section – beyond the current study, one other was conducted in myxoid liposarcomas, the second in marginally resectable soft tissue sarcomas.

  1. Would a chart review have been better?

The inclusion criteria were designed prospectively, all patients signed upfront informed consent, it is true that the study was designed before 2010 so methodology was very simple but still it was prospective study.

  1. The doses of RT were chosen on what basis?

The rationale for hypofractionation regimen was based on radiobiological grounds. Data is limited in terms of determination of alpha/beta ratio in sarcomas. Based on the results for liposarcomas published by Reitan and Kaalhus the a/b ratio was estimated at 0.4 (-1.4, 5.4)[1, 2]. To compare the total doses given to the patients with the use of various fraction doses in various treatment periods, we converted the physical doses into biologically equivalent doses (BEDs). Following formula proposed by Fowler[3] and modified for easy daily practice calculations was used: BED = ndapplied (dapplied + a/b)/(dreference + a/b) where n is the number of fractions, applied is the fraction size of the applied regimen, dreference is the conventional fraction size of 2 Gy, a/b is the ratio of radiation fractionation sensitivity (which has been assumed to be equal to 0.4 Gy for liposarcomas according to Reitan and Kaalhus). Using this formula, the BED was 56.25 Gy for 5x5 Gy scheme compared to 50 Gy in conventional schedule of 25x2Gy and 33 total treatment days. It was described earlier in our papers. IT HAS BEEN ADDED TO DISCUSSION SECTION.

  1. Reitan JB, Kaalhus O: Radiotherapy of liposarcomas. Br J Radiol 1980, 53(634):969-975.
  2. Thames HD, Suit HD: Tumor radioresponsiveness versus fractionation sensitivity. Int J Radiat Oncol Biol Phys 1986, 12(4):687-691.
  3. Fowler JF: Biological factors influencing optimum fractionation in radiation therapy. Acta Oncol 2001, 40(6):712-717.

The answers to these questions should be integrated into the text.

As required these answers were integrated into the manuscript text.

5.Just as there is a lack of toxicity data that I would expect to see in older patients. Furthermore, how did elderly patients tolerate the therapies?

We have no evidence on worse tolerance in older population. Age was not a factor having impact with higher rates of treatment complications. It is stated in Results section.

Have I guess the doses been reduced or omitted of doxo for example? Although well presented, the general data are few and little discussed.

In our study we did not record chemotherapy toxicity or dose reductions, so we lack this data, as the study was focused only on combination of hypofractionated radiotherapy and surgery.

  1. I don't understand the difference between the LRFS of 81% at 5 yrs and the DFS 46% at 5 yrs, shouldn't they coincide? Or are they calculated differently?

Thank you for this remark but it is clearly related to the definition of LRFS and DFS (LRFS time was calculated from the date of the start of preoperative RT to the date of the most recent follow-up (censored) or LR. DFS time was cal-culated from the date of the start of preoperative RT to the date of the most recent follow-up (censored), recurrence, or death) – DFS comprised in the calculations all disease relapses (including metastatic disease) and in high risk patients this difference may occur frequently.

Reviewer 3 Report

The authors present a prospective study of hypofractionated radiation therapy prior to surgery for STS of extremity or trunk and show good tolerance with a low rate of local relapse. The study is well presented but could use extensive editing of English Language. I have the following comments and questions:

1.) Why were extremity and truncal sarcomas grouped together? They are generally staged different and have different etiologies and requirements for aggressive resection.

2.) A little bit more detail on patient selection would be helpful--was this all patients treated during the dates indicated in the methods? 

3.) Please add more details on the following: what was the preoperative imaging? Were tumors re-imaged after radiation and prior to XRT? How were patients staged systemically to rule out metastases prior radiation? What was the follow-up imaging scheme?
4.) On page 5 in first section of 'Results', you state the vast majority of tumors were on the 'tight'--I assume that should be 'thigh'?

5.) In those patients treated with chemotherapy, did you notice any impact on tumor response and/or recurrence?
6.) In Table 1, next to 5-year survival in the column headers, you should indicate the numbers are % (or median).

7.) In Table 2, it might be less confusing to have a separate column for the p-value as opposed to right next to the HR.

8.) The way you express 95% confidence interval in Table 3 is different than in Tables 1 and 2, and is confusing. You should be consistent from table to table. Also, within the results text, you refer to Table 3 as (Table nr 3)--not sure why the nr is there.

9.) In Table 4, you have 'inflammation' requiring oral antibiotics--assume you mean 'infection'?
10.) In the discussion, last sentence of 2nd paragraph (lines 285-287), it makes it sound like the local recurrence rate is >80% which is not true--the sentence should probably be re-written.

Author Response

The authors present a prospective study of hypofractionated radiation therapy prior to surgery for STS of extremity or trunk and show good tolerance with a low rate of local relapse. The study is well presented but could use extensive editing of English Language.

The manuscript has been corrected accordingly with English native speaker linguistic assistance.

I have the following comments and questions:

Why were extremity and truncal sarcomas grouped together? They are generally staged different and have different etiologies and requirements for aggressive resection.

We included in the study extremity and trank wall only sarcomas (not truncal sarcomas of the internal organs or thorax/abdomen/retroperitoneum), trunk wall and extremity are commonly grouped in the studies (and staging classification), as one of the pivotal studies: Gronchi A, Palmerini E, Quagliuolo V, Martin Broto J, Lopez Pousa A, Grignani G, Brunello A, Blay JY, Tendero O, Diaz Beveridge R, Ferraresi V, Lugowska I, Merlo DF, Fontana V, Marchesi E, Braglia L, Donati DM, Palassini E, Bianchi G, Marrari A, Morosi C, Stacchiotti S, Bagué S, Coindre JM, Dei Tos AP, Picci P, Bruzzi P, Casali PG. Neoadjuvant Chemotherapy in High-Risk Soft Tissue Sarcomas: Final Results of a Randomized Trial From Italian (ISG), Spanish (GEIS), French (FSG), and Polish (PSG) Sarcoma Groups. J Clin Oncol. 2020 Jul 1;38(19):2178-2186. doi: 10.1200/JCO.19.03289. Epub 2020 May 18. PMID: 32421444.

A little bit more detail on patient selection would be helpful--was this all patients treated during the dates indicated in the methods?

In the study were include all patients diagnosed with STS, regardless of previous treatment. In this analysis, we include only those patients with primary tumors.

Please add more details on the following: what was the preoperative imaging? Were tumors re-imaged after radiation and prior to XRT? How were patients staged systemically to rule out metastases prior radiation? What was the follow-up imaging scheme?

Imaging prior to the treatment was not exactly defined in the trial as it was done according to general recommendation in Polish guidelines. All patients as required in the guidelines had a CT scan of the thorax to exclude metastatic disease prior to treatment. We did not perform reimaging (CT or MRI) prior to surgery as the surgery was performed in few days after radiotherapy. During follow-up patients (according to ESMO guidelines) are assessed every 3 months in the first 2 years, then twice a year up to the fifth year, and once a year thereafter. Alternate ultrasound (US) and MRI of the scar are performed, as well as CT of the chest abdomen and pelvis, alternating with X-ray of the chest and US of the abdomen.

 4.) On page 5 in first section of 'Results', you state the vast majority of tumors were on the 'tight'--I assume that should be 'thigh'?

Of course, this spelling error was corrected.

5.) In those patients treated with chemotherapy, did you notice any impact on tumor response and/or recurrence?

In our study, we did not analyze the treatment response assessed by imaging or pathological examination, as in majority of patients (those without chemotherapy) the whole treatment time was too short to induce a response, which we describe in the discussion as a potential disadvantage of our treatment method. Unfortunately, we did not see any impact on chemotherapy in terms of improving overall survival but the use of chemotherapy was usually biased by higher risk tumors.

 6.) In Table 1, next to 5-year survival in the column headers, you should indicate the numbers are % (or median).

It has been changed.

7.) In Table 2, it might be less confusing to have a separate column for the p-value as opposed to right next to the HR.

We have changed the presentation of data in more clear way.

8.) The way you express 95% confidence interval in Table 3 is different than in Tables 1 and 2, and is confusing. You should be consistent from table to table.

The Table was changed as required

9.) Also, within the results text, you refer to Table 3 as (Table nr 3)--not sure why the nr is there.

It has been changed

10.) In Table 4, you have 'inflammation' requiring oral antibiotics--assume you mean 'infection'?

It has been changed

11.) In the discussion, last sentence of 2nd paragraph (lines 285-287), it makes it sound like the local recurrence rate is >80% which is not true--the sentence should probably be re-written.

Thank you for this comment, of course the sentence was changed.

Round 2

Reviewer 1 Report

All my concerns have been adressed adequately, thus I recommend this manuscript for publication.

Author Response

Thank you very much for your help and valuable comments

Reviewer 2 Report

Dear Authors,

thank you for improuving the manuscript, to me not suitable for pubblication yet for these:

patients are not homegeneous (some had neoadjuvant CT and some any)

some data ie median LR is not reached

population bias is relevant 

impact of CT is not underlayed mainly in SS where CT showed impact on survival in young adults < 45yrs of age

I.e. Response to chemotherapy was 55.2%, including 22.4% cases with complete or major partial remissions, and 32.8% with minor partial remissions. Ferrari A et al.

These bias effected the survival indeed, loosin efficacy.

Considering this, conclusion could be relevant for toxicity but not for survival and prognosis as you stated in conclusions.

I would suggest to consider to extrapooled the populatoin and LRFS distinguishing beetwen the two main population who had neodjuvant and adjuvant, who only adjuvant who none. Who had R0 and R1 giving separete survival data. It would be better for readers.

Sincerely,

Author Response

Thank you for this comment; we believe that responding to this issue greatly improved our paper. Indeed, in the multivariable model, we did not show differences between patients who received chemotherapy (CHTH+) and those who did not (CHTH-). We would like to underline that the trial design was focused on preoperative hypofractionated radiotherapy and we did not exclude perioperative chemotherapy as exclusion criteria. This may be due to several factors. First, our research was focused on radiotherapy treatment. This study was neither designed nor powered to show differences in survival in those two groups.  Secondly, as we showed in the newly added table 3., the CHTH+/CHTH- groups were not balanced. There may be a residual confounding present in the multivariable models, which influences the results. Lastly, this study extends over eight years.  At this time, several chemotherapy regimens were used. Possibly, not all of them were optimal and visibly affected OS of the patients. At the same time, the HR (0.85) coefficient in the model in table 4 suggests benefit in chemotherapy patients and a 15% reduction in risk of death. The effect is not significant due to the factors mentioned earlier.

To reiterate. In order to respond to this remark, we have added table 3. To visualize differences in groups and passage in the discussion to explain this issue.

Reviewer 3 Report

The majority of comments have been addressed

Author Response

Thank you very much for your help and valuable comments.

Round 3

Reviewer 2 Report

In this form I endorse for pubblication

Thank you the authors for their effort

Author Response

(The authors gave the same response as above.)
